# VerityMath: Advancing Mathematical Reasoning by Self-Verification Through Unit Consistency

Vernon Toh Yan Han [1]    Ratish Puduppully [2]    Nancy F. Chen [2 3 4]

## Abstract

Large Language Models (LLMs), combined with program-based solving techniques, are increasingly demonstrating proficiency in mathematical reasoning. For example, closed-source models such as OpenAI GPT-4 and Claude show excellent results in solving math word problems. However, progress in math word problem-solving for open-source LLMs is limited, and the challenges these models face are not well-studied. In this paper, we study the performance of strong open-source LLMs, including Llama 2 (7B), Code Llama (7B), and Mistral (7B) on math word problems using program-based solving techniques. Specifically, we analyze the outputs of these models when applied to math word problems and identify a category of problems that pose a significant challenge, particularly those involving quantities spanning multiple units. To address this issue, we propose a systematic approach by defining the units for each quantity and ensuring the consistency of these units during mathematical operations. We developed *Unit Consistency Programs* (UCPs), an annotated dataset of math word problems, each paired with programs containing unit specifications and unit verification routines. We fine-tuned Llama 2 (7B), Code Llama (7B), and Mistral (7B) models with UCPs to produce their *VerityMath* variants. Our findings indicate that our approach, which incorporates unit consistency, currently slightly underperforms compared to an approach that does not. To understand the reasons behind this, we conduct an in-depth error analy-

sis and suggest options for future improvements. Our code and dataset are available at https://github.com/vernontoh/VerityMath.

## 1. Introduction

The ability to reason during the process of thinking and decision-making is a fundamental aspect of human intelligence. Replicating this ability in machines has been an objective in the field of Natural Language Processing. Large language models (LLMs) (OpenAI, 2023; Anil et al., 2023) mark significant progress toward this goal, demonstrating remarkable proficiency across a range of tasks, including mathematical reasoning (Zhou et al., 2023; Zhao et al., 2023; Zheng et al., 2023). Specifically, methods like Program Aided Language Model (PAL) (Gao et al., 2023) as well as Program of Thoughts (PoT) (Chen et al., 2023) have demonstrated improvements in LLMs' ability to solve complex mathematical problems. These methodologies empower LLMs to formulate programs as intermediate reasoning steps and delegate the execution of these steps to a Python interpreter, thereby enhancing computational accuracy.

However, open-source LLMs like those referenced in (Touvron et al., 2023; Rozière et al., 2023; Jiang et al., 2023) demonstrate limited success in math reasoning tasks. For example, after fine-tuning on the GSM8K-PAL dataset provided by Jie & Lu (2023), Mistral (7B) achieves just 70.4% accuracy on GSM8K (Cobbe et al., 2021) (Ref Table 4). Our analysis of the fine-tuned Llama 2 (7B), Code Llama (7B) and Mistral (7B) reveals challenges in solving math word problems with multi-unit quantities. These issues are more pronounced in multi-step reasoning, where early errors can lead to incorrect final solutions. Our study thus identifies specific challenges the model faces.

We propose a methodological framework to enhance the reasoning capabilities of LLMs by introducing a unit system for quantities and enforcing unit consistency. Ensuring unit consistency is crucial for accurate solutions in the context of mathematical word problems. To achieve this, we introduce Unit Consistency Programs (UCPs) (Figure 1) designed to enhance LLMs' reasoning abilities by enabling them to self-verify unit consistency within equations. UCPs consist of

---

*Most of the work was done while the first author was an intern at A*STAR. [1]Singapore University of Technology and Design [2]Institute for Infocomm Research (I²R), A*STAR, Singapore [3]CNRS@CREATE, Singapore [4]Centre for Frontier AI Research (CFAR), A*STAR, Singapore. Correspondence to: Vernon Toh Yan Han <vernon_toh@mymail.sutd.edu.sg>, Ratish Puduppully <puduppully@i2r.a-star.edu.sg>.

*The first AI for MATH Workshop at the 41st International Conference on Machine Learning*, Vienna, Austria. Copyright 2024 by the author(s).

**Question: Davos bought 2 shirts from the mall that cost $30 each. If he bought them with a 40% discount, how much did Davos pay for the 2 shirts?**

**Model Output** — **PAL Program**

```
def solution():
    shirt_cost = 30
    num_shirts = 2
    discount_percent = 40
    discount_amount = shirt_cost * discount_percent / 100
    total_cost = shirt_cost * num_shirts - discount_amount
    result = total_cost
    return result
```

**Inconsistent Units**

$$dollars = \left( \frac{dollars}{shirts} \times shirts \right) - \left( \frac{dollars}{shirts} \right)$$

**Model Output** — **Unit Consistency Program**

```
def solution():
    shirts_count = 2
    shirts_count_unit = Counter({"shirts": 1})
    cost_per_shirt = 30
    cost_per_shirt_unit = Counter({"dollars": 1, "shirts": -1})
    total_cost_before_discount = shirts_count * cost_per_shirt
    total_cost_before_discount_unit = Counter({"dollars": 1})
    assert total_cost_before_discount_unit == shirts_count_unit + cost_per_shirt_unit
    discount_rate = 0.40
    discount_percentage_unit = Counter()
    discount_amount = total_cost_before_discount * discount_rate
    discount_amount_unit = Counter({"dollars": 1})
    assert discount_amount_unit == total_cost_before_discount_unit
    total_cost_after_discount = total_cost_before_discount - discount_amount
    total_cost_after_discount_unit = Counter({"dollars": 1})
    assert total_cost_after_discount_unit == total_cost_before_discount_unit == discount_amount_unit
    result = total_cost_after_discount
    return result
```

**Units**

$$shirts^1$$

**Units**

$$\frac{dollars}{shirts} = dollars^1 \times shirts^{-1}$$

**LHS Units**

$$dollars^1$$

$$=$$

**RHS Units**

$$(shirts^1) \times (dollars^1 \times shirts^{-1}) = dollars^1 \times shirts^{1 + (-1)}$$

Figure 1: Comparison between PAL-based Programs and Unit Consistency Programs. Unit Consistency Programs contain unit specifications using `Counter` objects and unit verification routines using `assert` statements.

`Counter` objects responsible for tracking variable units and `assert` statements generated following each equation involving an operation. These `assert` statements verify the consistency of units within the equation and can trigger an `assert` error when inconsistent units are detected.

We have developed a dataset that pairs math word problems with unit consistency programs containing unit specifications and verification routines. Our preliminary study presents the outcomes of fine-tuning Llama 2 (7B), Code Llama (7B) , and Mistral (7B) using these programs. Although our approach, which incorporates unit consistency, currently slightly underperforms compared to a non-unit-consistent approach, we conducted an in-depth error analysis to understand the reasons behind this discrepancy and proposed several options for future improvements.

## 2. Motivating Analysis

Recent studies have utilized the concept of program-based prompting to generate pseudo-gold programs as an intermediary step for training smaller models (Jie & Lu, 2023; Zhu et al., 2023). Notably, this approach has shown promising outcomes, although these results still fall short of the performance achieved by larger models like GPT-4 (OpenAI, 2023). To better comprehend the gaps in the mathematical reasoning abilities of smaller models, we fine-tuned Llama 2 (7B) (Touvron et al., 2023), Code Llama (7B) (Rozière et al., 2023), and Mistral (7B) (Jiang et al., 2023) using the GSM8K-PAL dataset provided by Jie & Lu (2023), and conducted a comprehensive analysis of the fine-tuned models. The GSM8K-PAL dataset contains approximately 6.8k

word problems paired with their PAL annotations in the training dataset as shown in Table 1.

After fine-tuning these models on GSM8K-PAL, we observed that they struggle with math word problems involving multiple different units. As illustrated in Figure 1 (top), the example illustrates a unit mismatch in the model trained on the PAL-based approach. Specifically, the subtraction operation between variables `shirt_cost` × `num_shirts` and `discount_amount` is incorrect. The units are incompatible: the former is in dollar, and the latter is in dollar/shirt.

To support our observation that the model struggles with problems containing multiple units, we employed GPT-3.5 Turbo[1] to categorize the examples from both the train and test splits into two distinct groups. The first group comprises of questions involving a single unit, while the second group comprises of questions with multiple units. This classification was achieved using few-shot prompting, with GPT-3.5 Turbo serving as the backend engine. The specifics of the few-shot prompt utilized are detailed in Appendix A.2, and the distribution of these categories is presented in Table 2. Our analysis reveals that approximately 40% of the problems in both training and test splits involve multiple units.

To further evaluate the accuracy of GPT-3.5 Turbo in identifying questions with multiple units, we conducted a small-scale human assessment, detailed in Table 3. The first author manually annotated 100 randomly selected test examples from GSM8K and compared the annotations with the classifications made by GPT-3.5 Turbo. The results demonstrated

---

[1]GPT-3.5 Turbo annotations were obtained in September 2023.

| Dataset | #Train | #Program | #Valid | #Test |
|---|---|---|---|---|
| GSM8K-PAL | 7,473 | 6,877 (92.0%) | - | 1,319 |
| UCPs | 7,473 | 4,480 (59.9%) | - | 1,319 |

Table 1: Comparison of dataset size of GSM8K-PAL by (Jie & Lu, 2023) and UCPs.

| Train Dataset (7473) | | Test Dataset (1319) | |
|---|---|---|---|
| Single | Multiple | Single | Multiple |
| 4479 | 2994 | 755 | 564 |
| (59.9%) | (40.1%) | (57.2%) | (42.8%) |

Table 2: Classification of GSM8K into two categories: single unit and multiple units.

| | Positive Predicted | Negative Predicted |
|---|---|---|
| **Actual Positive** | 37 | 16 |
| **Actual Negative** | 9 | 38 |
| **Precision** | **Recall** | **Accuracy** |
| 80.4% | 69.8% | 75.0% |

Table 3: Small human evaluation compared on GPT-3.5 Turbo classification on 100 randomly sampled test examples from GSM8K. Human annotations were done by the first author.

a precision of 80.4%, indicating that GPT-3.5 Turbo generally excels in predicting questions involving multiple units. We have extended this analysis to the SVAMP (Patel et al., 2021), as presented in Appendix A.3, to demonstrate that this phenomenon is not exclusive to GSM8K.

Based on the test dataset split we collected, we divided the accuracy of the fine-tuned models into two categories: one for questions with a single unit and another for questions with multiple units. This categorization is shown in Table 4. A detailed examination of Table 4 reveals that our observations remained consistent across all three fine-tuned models, indicating superior performance on single-unit problems compared to those with multiple units. Motivated by these findings, we developed Unit Consistency Programs (UCPs) aimed at addressing the limitations inherent in PAL-based solutions.

# 3. Methodology

## 3.1. Unit Consistency Programs

Unit consistency checks are essential safeguards, helping to identify and prevent errors from inconsistent units in mathematical equations. In contrast to PAL/PoT approaches that directly generate programs to solve math word problems, our method enhances these programs by integrating specialized `Counter` objects. These objects are responsible for tracking variable units and ensuring the correct handling of operations with differing units. Additionally, we incorporate `assert` statements after each equation, as illustrated in Figure 1 (bottom). These `assert` statements verify unit consistency within equations, triggering an error if unit mismatches are detected.

Consider the example in Figure 1 (bottom), illustrating a multiplication operation between `shirts_count` (measured in 'shirts') and `cost_per_shirt` (measured in 'dollars per shirt'). In this operation, the units of 'shirts' from

`shirts_count` and 'per shirt' from `cost_per_shirt` naturally cancel each other out, resulting in a unit of 'dollars'. An `assert` statement is used to verify this expected cancellation of units. In our notation, the exponent of a unit in the numerator is represented as +1, and in the denominator as -1. Therefore, in this multiplication, the positive exponent of 'shirts' in `shirts_count` cancels with the negative exponent of 'per shirt' in `cost_per_shirt`, aligning the product's right-hand side (RHS) with the expected left-hand side (LHS) unit of `total_cost_before_discount`, confirming it is in 'dollars'. The example also illustrates a unitless quantity, specifically a percentage. In this case, there won't be any units specified in the `Counter` initialization. Our methodology requires the development of a specialized `Counter` class, details of which are elaborated in the Appendix A.4.2.

## 3.2. Training Data Annotations

Adopting the methodology used in PAL/PoT, we sampled programs for each math word problem, adding them to our training data when their execution yielded the correct answer. For each math word problem $x$ in the training dataset $D$, we performed greedy decoding at temperature $T = 0$ to synthesize program $P_{ucp}$. Upon executing the program $P_{ucp}$, if the predicted answer $\hat{y}$ matched the ground-truth answer $y$ and $P_{ucp}$ consists of `Counter` objects and `assert` statements, we included the tuple $(x, P_{ucp}, y)$ in our new training dataset $D_{ucp}$. Any math word problem $x$ for which a matching program couldn't be obtained was discarded.

## 3.3. Fine-tuning Small Models

We fine-tuned smaller models with our annotated dataset $D_{ucp}$ through standard causal language modeling techniques. The objective is to generate a corresponding Python program $P$ for a given math word problem $x$. After fine-tuning, the model was used to generate Python programs, which were then executed using a Python interpreter to obtain the final answer. We employed strong open-source LLMs such as Llama 2 (7B), Code Llama (7B), and Mistral (7B) as our models to fine-tune.

| Model | Single | Multiple | Overall |
|---|---|---|---|
| *Closed-Source Models* | | | |
| GPT-4 | - | - | 92.0 |
| GPT-3.5-Turbo | - | - | 80.8 |
| *Open-Source Models 7B* | | | |
| Llama-2 (PAL)[†] | 58.5 ↑3.1 | 51.2 ↓4.2 | 55.4 |
| Code-Llama (PAL)[†] | 65.6 ↑2.5 | 59.8 ↓3.3 | 63.1 |
| Mistral (PAL)[†] | 72.2 ↑1.8 | 68.1 ↓2.3 | 70.4 |
| VerityMath-Llama-2 | 51.9 ↑5.7 | 38.7 ↓7.5 | 46.2 |
| VerityMath-Code-Llama | 58.4 ↑4.2 | 48.6 ↓5.6 | 54.2 |
| VerityMath-Mistral | 71.5 ↑3.3 | 63.7 ↓4.5 | 68.2 |

Table 4: Comparison of test accuracy on GSM8K of different 7B open-source models fine-tuned on PAL and UCP. The green and red boxes represent the increase and decrease in accuracy compared to its overall score. [†]We fine-tune the model using GSM8K-PAL by Jie & Lu (2023).

## 4. Experiments

### 4.1. Dataset

We conducted our experiments primarily on GSM8K, employing few-shot prompting with GPT-4 for the first 1,000 examples[2] and GPT-4 Turbo for the remaining 6,473 examples[3] in the GSM8K train dataset. We used six manually crafted Unit Consistency Programs (UCPs) samples, as detailed in Appendix A.1. We successfully annotated 59.9% of the GSM8K train dataset, creating our annotated UCPs dataset, $D_{ucp}$. Table 1 presents the statistics of our UCPs dataset.

### 4.2. Baseline

Our baseline models consist of different models such as Llama 2 (7B), Code Llama (7B), and Mistral (7B) fine-tuned on GSM8K-PAL. We use this as a direct baseline to our method as it provides a more effective comparison between our method UCPs and existing methods like PAL/POT since our UCPs serve as extensions to typical Python programs used for solving mathematical problems, as demonstrated in PAL/POT.

### 4.3. Implementation

We conducted fine-tuning experiments on GSM8K-PAL and UCPs, details of both datasets can be found in Table 1. In our fine-tuning experiments, we utilized the QLoRA technique (Dettmers et al., 2023) for enabling efficient fine-tuning. All QLoRA hyper-parameters were set as presented in Dettmers et al. (2023). In all our experiments we use NF4 with double quantization and bf16 computation datatype. We set LoRA

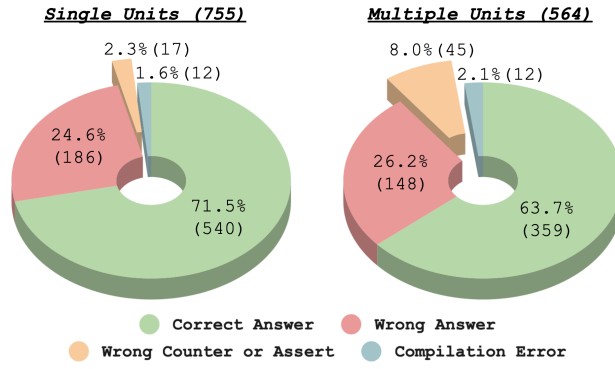

Figure 2: Error analysis of VerityMath-Mistral (7B). Correct Answer: The program compiles and produces the correct answer. Wrong Answer: The program compiles but produces an incorrect answer. Wrong `Counter` or `assert`: After removing `Counter` and `assert` statements, the program produces the correct answer. Compilation Error: The program is unable to compile.

$r = 64$, $\alpha = 16$, and add LoRA modules on all linear layers of the base model. We also use max grad norm of 0.3 and LoRA dropout of 0.1. We use AdamW optimizer and set the learning rate to $2e^{-4}$, with a batch size of 32 and a maximum context length of 1024. We trained the model for 20 epochs using 4 A100 40 GB GPUs which took roughly 14 hours and evaluated it on the test dataset.

### 4.4. Main Results

Our model, VerityMath-Mistral (7B), fine-tuned on UCPs achieved an overall accuracy of 68.2% on the GSM8K test dataset. Specifically, it attained 71.5% accuracy for problems involving a *single* unit and 63.7% accuracy for those with *multiple* units, as detailed in Table 4. When compared to the Mistral (7B) (PAL) baseline, VerityMath-Mistral (7B) exhibits a slight overall accuracy decrease of 2.2%. Meanwhile, VerityMath-Code Llama (7B) and VerityMath-Llama 2 (7B) experienced more significant declines in their overall accuracy, approximately 9% lower than their respective PAL counterparts. Specifically, VerityMath-Code-Llama achieved 54.2% overall accuracy, with 58.4% for single unit problems and 48.6% for multiple units, while VerityMath-Llama-2 achieved an overall accuracy of 46.2%, with 51.9% for single unit and 38.7% for multiple units.

### 4.5. Analysis

In this section, we conducted an in-depth analysis of the potential causes for the decline in overall accuracy in the GSM8K test dataset. We focused on VerityMath-Mistral (7B) for all of our analysis.

---

[2]GPT-4 annotations obtained in September 2023.

[3]GPT-4 Turbo annotations obtained in December 2023.

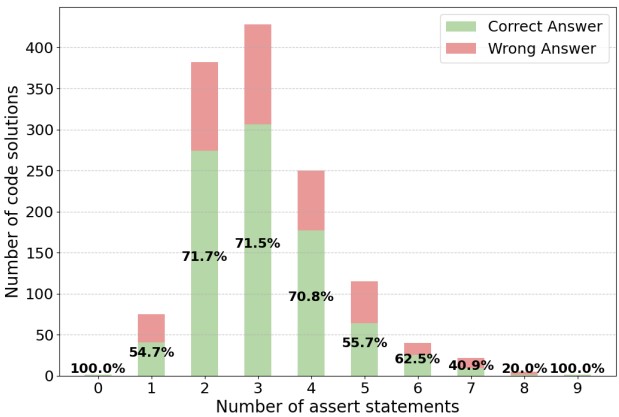

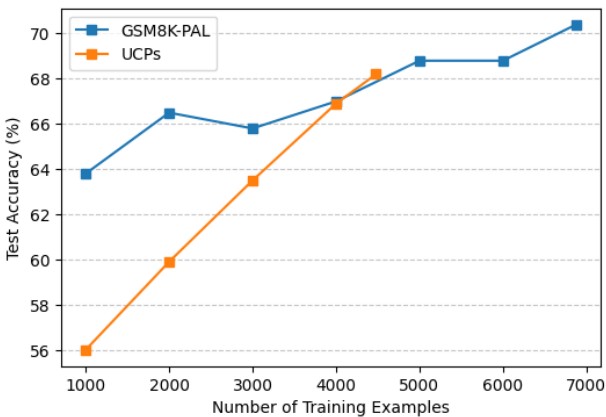

Figure 3: Performance of VerityMath-Mistral (7B) on the GSM8K test dataset based on the number of `assert` statements in the code solution. The percentage shown in each bar represents the percentage of correct answers given the number of `assert` statements in the code solution.

Figure 4: Performance of VerityMath-Mistral (7B) as we scale the number of training examples of GSM8K-PAL and UCPs. GSM8K-PAL has a total of 6877 annotated training examples whereas UCPs have 4480 annotated training examples.

**Correctness of `Counter` and `assert` statements** In an error analysis of VerityMath-Mistral (7B) outputs from the test dataset, we observed some challenges that led to decreased performance, specifically, the correctness of `Counter` and `assert` statements. We reran the whole evaluation but this time, when we were met with a program that raised an assertion error, we removed the `Counter` and `assert` statements and executed the programs again. If the program compiles and produces the correct answer after this modification, it indicates that the program was originally incorrect due to incorrect `Counter` or `assert` statements. Referring to Figure 2, we observed a notable percentage of output programs that contained incorrect `Counter` or `assert` statements in VerityMath-Mistral (7B) outputs. Specifically, 2.3% of the problems with single units and 8.0% of the problems with multiple units have incorrect `Counter` and `assert` which caused correct programs that would have resulted in the correct answer to have a false assertion error resulting in the wrong answer. Examples of such cases with incorrect `Counter` and `assert` are shown in Appendix A.5.2.

**Performance based on number of `assert` statements** We further conducted a detailed analysis of code solutions categorized by the number of assert statements, as shown in Figure 3. Each bar represents the total number of code solution that consists of a specific number of assert statements. The green segments of the bars indicate the count of code solutions that resulted in the correct answer, while the red segments represent those that resulted in an incorrect answer. The percentage of correct answers is annotated on each bar for clarity. It is evident from the plot that the percentage of correct answers generally decreases as the number of assert

statements increases, from code solutions with 2 to 4 assert statements having approximately 70% accuracy to code solutions with 5, 6, and 7 assert statements having 55.7%, 62.5%, and 40.9% respectively. Highlighting a trend where more complex code solutions with more assert statements are more likely to result in incorrect answers. This aligns with the earlier observations regarding the correctness of `assert` statements, and suggests that with more assert statements in the code solution, it is more prone to having errors due to the incorrect `assert` statements which would then result in a wrong answer.

**Impact of training annotations** Due to the difference in the number of training examples between GSM8k-PAL and UCPs of 2397 as shown in Table 1. It is crucial to also understand the implications of the number of training examples with respect to the performance. We fine-tuned Mistral (7B) on both GSM8k-PAL and UCPs with an interval of 1000 training examples and showed the results in Figure 4. The performance of Mistral (7B) when fine-tuned on GSM8k-PAL or UCPs demonstrates a clear trend of improvement with the increase in the number of training examples. For GSM8k-PAL, the test accuracy starts at 63.8% with 1,000 training examples and steadily increases to 70.4% with 6,877 examples. On the other hand, The UCPs exhibit a more pronounced improvement curve, starting at 56.0% accuracy with 1,000 training examples, the performance increases significantly to 68.2% with 4,480 examples. This rate of improvement indicates that with limited examples, the concept of UCPs is harder to grasp for Mistral (7B) as compared to PAL. The difference in performance gains suggests that UCPs might have untapped potential that could be realized with an increased number of training ex-

amples and it implies that with sufficient training examples, UCPs could potentially surpass PAL in performance.

In our in-depth anaysis, we identified a notable bottleneck in our current method, which is the correctness of `Counter` and `assert` statements. This issue led to a slight decrease in performance. Our method, UCPs, is a relatively more complex method for existing 7B LLMs to learn, but with a significant increase in dataset annotations, it is highly possible that our method will outshine the existing PAL method. Another approach could involve data augmentation using synthetic examples (Wu et al., 2021). Examples showcasing the efficacy of UCPs are available in Appendix A.5.1.

## 5. Related Work

Our research builds upon the Program of Thoughts (PoT) approach (Chen et al., 2023) and the Program Aided Language Model (PAL) (Gao et al., 2023) methodologies, which have shown effectiveness in solving mathematical problems. These approaches have outperformed techniques like the Chain-of-Thought (CoT) (Wei et al., 2022), which can struggle with computational inaccuracies (Lewkowycz et al., 2022). We extend their work by focusing on the use of programs for solving math word problems and the concept of self-verification to improve LLMs' reasoning capabilities.

The advancement of GPT models (Brown et al., 2020) has inspired various studies (Ho et al., 2023; Fu et al., 2023; Magister et al., 2023; Shridhar et al., 2023) on creating synthetic datasets for fine-tuning smaller models (Hinton et al., 2015). Notably, Zhu et al. (2023) used PAL annotations in this context, (Magister et al., 2023; Ho et al., 2023; Yu et al., 2023) employed CoT annotations, and Yue et al. (2023) used a hybrid of CoT and PoT rationales.

In mathematical problem-solving, ensuring solution validity is crucial due to hallucinations in LLMs (Bubeck et al., 2023) and challenges in executing multiplications (Dziri et al., 2023). Prior research has focused on training additional verifiers for answer accuracy (Cobbe et al., 2021), providing feedback for each intermediate reasoning step (Lightman et al., 2023), and integrating tools to agents(Gou et al., 2024). However, Weng et al. (2023) and Miao et al. (2023) have shown potential for LLMs to self-verify solutions. Our approach builds on these insights, incorporating programs for solving math word problems and leveraging self-verification to enhance LLM reasoning.

## 6. Conclusion and Future Work

In this study, we analyzed open-source Large Language Models (LLMs) and pinpointed their struggle with math problems involving multiple units, highlighting a key improvement area. We introduced Unit Consistency Programs (UCPs) as a novel method to address LLMs' reasoning and verification abilities, especially in complex math problems. We identified some limitations in our current approach. Future work will focus on advancing unit check methodologies in UCPs to address these limitations.

## Limitations

Recent creations of synthetic datasets for math problem-solving often rely on prompting large language models (LLMs), such as GPT-4. However, this approach can be costly, and on a large scale, the expenses escalate. Our dataset creation incurred a total cost of approximately $350 USD. Due to budget constraints, we couldn't sample multiple reasoning paths per question, as presented in Wang et al. (2023), limiting the potential for increased annotations.

## Impact Statement

This paper presents work whose goal is to advance the field of math problem-solving using LLMs. However, it is also crucial to be aware of the potential risks associated with VerityMath. Due to the current challenges of VerityMath, the units initialized by `Counter` and `assert` statements may not always be accurate. Consequently, it is strongly recommended to exercise caution when relying on VerityMath outputs for any use.

## Acknowledgements

This research is supported by the Ministry of Education, Singapore, under its Science of Learning Grant (award ID: MOESOL2021-0006). Any opinions, findings and conclusions or recommendations expressed in this material are those of the author(s) and do not reflect the views of the Ministry of Education, Singapore. The computational work for this article was partially performed on resources of the National Supercomputing Centre (NSCC), Singapore (https://www.nscc.sg).

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

# A. Appendix

## A.1. Manually constructed context samples for program synthesis

### A.1.1. 6-SHOT PROMPT

Question: Ashley bought a big bag of 96 candies. Yesterday, she ate 15 candies and today, she ate twice as much candies as yesterday. How many candies were left?

```python
Python solution:
def solution():
    """Ashley bought a big bag of 96 candies. Yesterday, she ate 15 candies and today, she ate twice as much candies
        as yesterday. How many candies were left?"""
    candies_initial = 96
    candies_initial_unit = Counter({"candies": 1})
    candies_ate_yesterday = 15
    candies_ate_yesterday_unit = Counter({"candies": 1})
    candies_ate_today = candies_ate_yesterday * 2
    candies_ate_today_unit = Counter({"candies": 1})
    assert candies_ate_today_unit == candies_ate_yesterday_unit
    total_candies_eaten = candies_ate_yesterday + candies_ate_today
    total_candies_eaten_unit = Counter({"candies": 1})
    remaining_candies = candies_initial - total_candies_eaten
    remaining_candies_unit = Counter({"candies": 1})
    assert remaining_candies_unit == candies_initial_unit == total_candies_eaten_unit
    result = remaining_candies
    return result
```

Question: There are 235 books in the library. On Monday, 122 books were taken out. On Tuesday, half of the books taken on Monday were brought back. How many books are there now?

```python
Python solution:
def solution():
    """There are 235 books in the library. On Monday, 122 books were taken out. On Tuesday, half of the books taken
        on Monday were brought back. How many books are there now?"""
    books_initial = 235
    books_initial_unit = Counter({"books": 1})
    books_taken_monday = 122
    books_taken_monday_unit = Counter({"books": 1})
    books_remaining_after_monday = books_initial - books_taken_monday
    books_remaining_after_monday_unit = Counter({"books": 1})
    assert books_remaining_after_monday_unit == books_initial_unit == books_taken_monday_unit
    books_brought_back_tuesday = books_taken_monday / 2
    books_brought_back_tuesday_unit = Counter({"books": 1})
    assert books_brought_back_tuesday_unit == books_taken_monday_unit
    books_remaining_after_tuesday = books_remaining_after_monday + books_brought_back_tuesday
    books_remaining_after_tuesday_unit = Counter({"books": 1})
    assert books_remaining_after_tuesday_unit == books_remaining_after_monday_unit ==
        books_brought_back_tuesday_unit
    result = books_remaining_after_tuesday
    return result
```

Question: There is a group of 10 people who are ordering pizza. If each person gets 2 slices and each pizza has 4 slices, how many pizzas should they order?

```python
def solution():
    """There is a group of 10 people who are ordering pizza. If each person gets 2 slices and each pizza has 4
        slices, how many pizzas should they order?"""
    people_total = 10
    people_total_unit = Counter({"people": 1})
    pizza_slices_per_person = 2
    pizza_slices_per_person_unit = Counter({"slices": 1, "people": -1})
    pizza_slices_total = people_total * pizza_slices_per_person
    pizza_slices_total_unit = Counter({"slices": 1})
    assert pizza_slices_total_unit == people_total_unit + pizza_slices_per_person_unit
    slices_per_pizza = 4
    slices_per_pizza_unit = Counter({"slices": 1, "pizza": -1})
    pizza_total = pizza_slices_total / slices_per_pizza
    pizza_total_unit = Counter({"pizza": 1})
    assert pizza_total_unit == pizza_slices_total_unit - slices_per_pizza_unit
    result = pizza_total
    return result
```

Question: Lana has 2 bags with 2 marbles in each bag. Markus has 2 bags with 3 marbles in each bag. How many more marbles does Markus have?

```python
Python solution:
def solution():
    """Lana has 2 bags with 2 marbles in each bag. Markus has 2 bags with 3 marbles in each bag. How many more
        marbles does Markus have?"""
```

```
bags_lana = 2
bags_lana_unit = Counter({"bags": 1})
marbles_per_bag_lana = 2
marbles_per_bag_lana_unit = Counter({"marbles": 1, "bags": -1})
marbles_total_lana = bags_lana * marbles_per_bag_lana
marbles_total_lana_unit = Counter({"marbles": 1})
assert marbles_total_lana_unit == marbles_per_bag_lana_unit + bags_lana_unit
bags_markus = 2
bags_markus_unit = Counter({"bags": 1})
marbles_per_bag_markus = 3
marbles_per_bag_markus_unit = Counter({"marbles": 1, "bags": -1})
marbles_total_markus = bags_markus * marbles_per_bag_markus
marbles_total_markus_unit = Counter({"marbles": 1})
assert marbles_total_markus_unit == marbles_per_bag_markus_unit + bags_markus_unit
marbles_more_markus = marbles_total_markus - marbles_total_lana
marbles_more_markus_unit = Counter({"marbles": 1})
assert marbles_more_markus_unit == marbles_more_markus_unit == marbles_total_lana_unit
result = marbles_more_markus
return result
```

Question: Sally has 4 containers with the same amount of cookies in them, totaling 12 cookies. John has 4 containers with the same amount of cookies in them, totaling 24 cookies. How many more cookies does John have in each container?

Python solution:
```
def solution():
    """Sally has 4 containers with the same amount of cookies in them, totaling 12 cookies. John has 4 containers
        with the same amount of cookies in them, totaling 24 cookies. How many more cookies does John have in each
        container?"""
    containers_sally = 4
    containers_sally_unit = Counter({"containers": 1})
    total_cookies_sally = 12
    total_cookies_sally_unit = Counter({"cookies": 1})
    cookies_per_container_sally = total_cookies_sally / containers_sally
    cookies_per_container_sally_unit = Counter({"cookies": 1, "containers": -1})
    assert cookies_per_container_sally_unit == total_cookies_sally_unit - containers_sally_unit
    containers_john = 4
    containers_john_unit = Counter({"containers": 1})
    total_cookies_john = 24
    total_cookies_john_unit = Counter({"cookies": 1})
    cookies_per_container_john = total_cookies_john / containers_john
    cookies_per_container_john_unit = Counter({"cookies": 1, "containers": -1})
    assert cookies_per_container_john_unit == total_cookies_john_unit - containers_john_unit
    more_cookies_per_container_john = cookies_per_container_john - cookies_per_container_sally
    more_cookies_per_container_john_unit = Counter({"cookies": 1, "containers": -1})
    assert more_cookies_per_container_john_unit == cookies_per_container_john_unit ==
        cookies_per_container_sally_unit
    result = more_cookies_per_container_john
    return result
```

Question: It takes Peter 4 hours to fix his fence. He spends 20 minutes everyday fixing the fence, how many days will it take for Peter to finish fixing the fence?

Python solution:
```
def solution():
    """It takes Peter 4 hours to fix his fence. He spends 20 minutes everyday fixing the fence, how many days will
        it take for Peter to finish fixing the fence?"""
    hours_to_fix_fence = 4
    hours_to_fix_fence_unit = Counter({"hours": 1})
    minutes_per_hour = 60
    minutes_per_hour_unit = Counter({"minutes": 1, "hours": -1})
    minutes_to_fix_fence = hours_to_fix_fence * minutes_per_hour
    minutes_to_fix_fence_unit = Counter({"minutes": 1})
    assert minutes_to_fix_fence_unit == hours_to_fix_fence_unit + minutes_per_hour_unit
    minutes_per_day_to_fix_fence = 20
    minutes_per_day_to_fix_fence_unit = Counter({"minutes": 1, "days": -1})
    total_days_to_fix_fence = minutes_to_fix_fence / minutes_per_day_to_fix_fence
    total_days_to_fix_fence_unit = Counter({"days": 1})
    assert total_days_to_fix_fence_unit == minutes_to_fix_fence_unit - minutes_per_day_to_fix_fence_unit
    result = total_days_to_fix_fence
    return result
```

### A.1.2. SYSTEM PROMPT

You are an experienced professional skilled in using python programs to solve math problems and enhancing code robustness through the strategic implementation of unit counters and assert statements to ensure flawless unit-level consistency checks. You will only write code blocks.

**A.2. Manually constructed context samples for Classification (Single Unit/ Multiple Units)**

A.2.1. 8-SHOT PROMPT

**Question**: In ten years, Samantha will be three times as old as Alex is today. If Alex is currently 14 years old, how old is Samantha now?

**Explanation**:

In this problem, a unit consistency check is not required to ensure the accuracy of measurements and calculations. We are dealing with a single unit - age (years). The calculations involve simple addition, subtraction, and multiplication of the same unit, so there is no need for unit conversions or comparisons between different units. In this case, all the quantities are given in terms of years, and there are no conflicting units or measurements that would require a unit consistency check. Answer: No.

**Question**: Sarah has a collection of stickers. Every alternate day, she adds 3 new stickers to her collection for 4 weeks. For each sticker, she spends 5 minutes sticking it onto her collection book. How many minutes did she spend sticking stickers?

**Explanation**:

In this problem, a unit consistency check is required to ensure the accuracy of measurements and calculations. We are dealing with different units - the number of days, the number of stickers, and the time spent sticking each sticker (minutes). By performing a unit consistency check, we can ensure that the conversions between different units are accurate and that the final result makes sense in the context of the problem. In this case, we can calculate the total number of stickers added by multiplying the number of stickers added per day by the number of days, and then calculate the total time spent sticking stickers by multiplying the number of stickers by the time spent sticking each sticker. Answer: Yes.

**Question**: Suzie practices piano every weekday for 3 weeks. Each day, she practices 2 songs that are 4 minutes long and 1 song that is 3 times longer than the other two songs combined. How many minutes does Suzie practice in total during these 3 weeks?

**Explanation**:

In this problem, a unit consistency check is required to ensure the accuracy of measurements and calculations. We are dealing with different units - the number of days, the number of songs, and the time spent practicing each song (minutes). By performing a unit consistency check, we can ensure that the conversions between different units are accurate and that the final result makes sense in the context of the problem. In this case, we can calculate the total time spent practicing each day by adding the time spent practicing the two shorter songs and the longer song, and then calculate the total time spent practicing over the three weeks by multiplying the daily practice time by the number of days. Answer: Yes.

**Question**: A girl rabbit hops along a garden path and picks 8 carrots. She places them in a basket with 16 other carrots she has already collected. While on her way home, the girl rabbit accidentally drops 2 carrots. How many carrots are left in the basket?

**Explanation**:

In this problem, a unit consistency check is not required to ensure the accuracy of measurements and calculations. We are dealing with a single unit - the number of carrots. The calculations involve simple addition and subtraction of the same unit, so there is no need for unit conversions or comparisons between different units. In this case, all the quantities are given in terms of the number of carrots, and there are no conflicting units or measurements that would require a unit consistency check. Answer: No.

**Question**: A bottle of shampoo is 500ml and each hair wash uses 20 ml of shampoo. Each salon charges $3.00 for one hair wash. How much money does a salon make on a bottle of shampoo?

**Explanation**:

In this problem, a unit consistency check is required to ensure the accuracy of measurements and calculations. We are dealing with different units - the volume of shampoo (milliliters), the volume of shampoo used per wash (milliliters), and

the cost of each wash (dollars). By performing a unit consistency check, we can ensure that the conversions between different units are accurate and that the final result makes sense in the context of the problem. In this case, we can calculate the total number of hair washes per bottle by dividing the total volume of the bottle by the volume used per hair wash, and then calculate the total money made by multiplying the number of hair washes by the cost of each hair wash. Answer: Yes.

**Question**: Natalie and Liam have a total of 72 stickers. However, Natalie has half the number of stickers as compared to Liam. How many stickers does Liam have?

**Explanation**:

In this problem, a unit consistency check is not required to ensure the accuracy of measurements and calculations. We are dealing with a single unit - the number of stickers. The calculations involve simple addition, subtraction, multiplication, and division of the same unit, so there is no need for unit conversions or comparisons between different units. In this case, all the quantities are given in terms of the number of stickers, and there are no conflicting units or measurements that would require a unit consistency check. Answer: No.

**Question**: Emma makes $35,000 a year. She spends $12,000 on rent every year, $3,000 on groceries every year, and $4,500 on gas every year. If she wants to have at least $15,000 left, how much more money must she make each year?

**Explanation**:

In this problem, a unit consistency check is not required to ensure the accuracy of measurements and calculations. We are dealing with a single unit - money (dollars). The calculations involve simple addition and subtraction of the same unit, so there is no need for unit conversions or comparisons between different units. In this case, all the quantities are given in terms of dollars, and there are no conflicting units or measurements that would require a unit consistency check. Answer: No.

**Question**: Maria pays $35/hour for 4 hours per week of flute lessons and $22/hour for 6 hours a week of guitar lessons. How much more does she spend on guitar lessons than flute lessons in a year?

**Explanation**:

In this problem, a unit consistency check is required to ensure the accuracy of measurements and calculations. We are dealing with different units - the cost per hour (dollars/hour) and the number of hours per week. By performing a unit consistency check, we can ensure that the conversions between different units are accurate and that the final result makes sense in the context of the problem. In this case, we can calculate the total cost of flute lessons per week by multiplying the cost per hour by the number of hours, and then calculate the total cost of guitar lessons per week by multiplying the cost per hour by the number of hours. Finally, we can calculate the difference in cost between guitar and flute lessons in a year by subtracting the total cost of flute lessons from the total cost of guitar lessons and multiplying it by the number of weeks in a year. Answer: Yes.

A.2.2. SYSTEM PROMPT

Create a coherent explanation for the importance of performing unit consistency checks in various mathematical problems and identify whether a question requires verification of unit consistency. Be as detailed as possible and write your explanation in 1 paragraph and end with Answer: Yes / No.

## A.3. Breakdown of SVAMP dataset

| Train Dataset (192) | | Test Dataset (81) | |
|---|---|---|---|
| **Single** | **Multiple** | **Single** | **Multiple** |
| 102 | 90 | 34 | 47 |
| (53.1%) | (46.9%) | (42.0%) | (58.0%) |

Table 5: SVAMP Dataset split. We only considered the portion which has type Multiplication or Common-Division.

The SVAMP dataset comprises a total of 1000 examples, with 700 allocated to the train dataset and 300 to the test dataset. The dataset encompasses four problem types: subtraction, addition, common-division, and multiplication. However, our analysis focuses solely on multiplication and common-division, as problems involving only addition or subtraction are defined to only consist of a single unit. We can observe from 5 that 46.9% and 58% of the problems are classified as multiple units in the train and test dataset respectively.

## A.4. Details on Unit Consistency Programs

### A.4.1. USAGE OF COUNTER CLASS

```
Counter({"marbles": 1, "bags":−1}  # marbles per bag

Counter({"slices": 1, "pizza":−1}  # slices per pizza

Counter({"books"}: 1)  # books

Counter()  # percentage (unitless)
```

### A.4.2. UNDESIRABLE BEHAVIOR OF COLLECTIONS COUNTER

Our work necessitated the creation of a custom `Counter` class. In programming, a 'class' is a blueprint for creating objects with specific attributes and behaviors. The standard `Counter` class, part of Python's `collections` module-a collection of specialized data types-was not fully suitable for our needs. It had the limitation of only counting positive values, meaning it would not accurately reflect counts when dealing with negative or zero values. Therefore, we had to create our own custom `Counter` class to handle this limitation.

```
from collections import Counter

Counter({"miles": 1}) − Counter({"hour": 1})
>>> Counter({'miles': 1})

Desired Output >>> Counter({'miles': 1, "hour": −1})
```

### A.4.3. ASSERT STATEMENT FORMULATION

```
Formulation of assert statement for equations with multiplication:
pizza_slices_total = people_total ∗ pizza_slices_per_person
assert pizza_slices_total_unit == people_total_unit + pizza_slices_per_person_unit

Formulation of assert statement for equations with division:
total_cups_sugar = total_teaspoons_sugar / teaspoons_per_cup_sugar
assert total_cups_sugar_unit == total_teaspoons_sugar_unit − teaspoons_per_cup_sugar_unit

Formulation of assert statement for equations with addition:
total_money = allowance + extra_money
assert total_money_unit == allowance_unit == extra_money_unit

Formulation of assert statement for equations with subtraction:
apps_remaining_after_deletion = apps_initial − apps_deleted
assert apps_remaining_after_deletion_unit == apps_initial_unit == apps_deleted_unit
```

## A.5. Outputs from VerityMath

### A.5.1. CORRECT OUTPUTS

```
def solution():
    """Kelly has 5 quarters and 2 dimes. If she buys a can of pop for 55 cents, how many cents will she have left
        ?"""
    quarters = 5
    quarters_unit = Counter({"quarters": 1})
    dimes = 2
    dimes_unit = Counter({"dimes": 1})
    cents_per_quarter = 25
    cents_per_quarter_unit = Counter({"cents": 1, "quarters": −1})
    cents_per_dime = 10
    cents_per_dime_unit = Counter({"cents": 1, "dimes": −1})
    total_cents_quarters = quarters ∗ cents_per_quarter
    total_cents_quarters_unit = Counter({"cents": 1})
    assert total_cents_quarters_unit == quarters_unit + cents_per_quarter_unit
    total_cents_dimes = dimes ∗ cents_per_dime
```

```
        total_cents_dimes_unit = Counter({"cents": 1})
        assert total_cents_dimes_unit == dimes_unit + cents_per_dime_unit
        total_cents = total_cents_quarters + total_cents_dimes
        total_cents_unit = Counter({"cents": 1})
        assert total_cents_unit == total_cents_quarters_unit == total_cents_dimes_unit
        cents_needed_for_pop = 55
        cents_needed_for_pop_unit = Counter({"cents": 1})
        cents_left = total_cents - cents_needed_for_pop
        cents_left_unit = Counter({"cents": 1})
        assert cents_left_unit == total_cents_unit == cents_needed_for_pop_unit
        result = cents_left
        return result
```

=> Example 287 in the test dataset.

```
def solution():
    """Every day, Wendi feeds each of her chickens three cups of mixed chicken feed, containing seeds, mealworms and
        vegetables to help keep them healthy. She gives the chickens their feed in three separate meals. In the
        morning, she gives her flock of chickens 15 cups of feed. In the afternoon, she gives her chickens another
        25 cups of feed. How many cups of feed does she need to give her chickens in the final meal of the day if
        the size of Wendi's flock is 20 chickens?"""
    cups_of_feed_per_chicken = 3
    cups_of_feed_per_chicken_unit = Counter({"cups": 1, "chickens": -1})
    chickens_flock_size = 20
    chickens_flock_size_unit = Counter({"chickens": 1})
    total_cups_of_feed_first_meal = 15
    total_cups_of_feed_first_meal_unit = Counter({"cups": 1})
    total_cups_of_feed_second_meal = 25
    total_cups_of_feed_second_meal_unit = Counter({"cups": 1})
    total_cups_of_feed_given = total_cups_of_feed_first_meal + total_cups_of_feed_second_meal
    total_cups_of_feed_given_unit = Counter({"cups": 1})
    assert total_cups_of_feed_given_unit == total_cups_of_feed_first_meal_unit ==
        total_cups_of_feed_second_meal_unit
    remaining_cups_of_feed = (chickens_flock_size * cups_of_feed_per_chicken) - total_cups_of_feed_given
    remaining_cups_of_feed_unit = Counter({"cups": 1})
    assert remaining_cups_of_feed_unit == chickens_flock_size_unit + cups_of_feed_per_chicken_unit
    result = remaining_cups_of_feed
    return result
```

=> Example 5 in test dataset.

```
def solution():
    """Tom's ship can travel at 10 miles per hour. He is sailing from 1 to 4 PM. He then travels back at a rate of 6
        mph. How long does it take him to get back?"""
    travel_rate_forward = 10
    travel_rate_forward_unit = Counter({"miles": 1, "hours": -1})
    time_forward = 4 - 1
    time_forward_unit = Counter({"hours": 1})
    distance_forward = travel_rate_forward * time_forward
    distance_forward_unit = Counter({"miles": 1})
    assert distance_forward_unit == travel_rate_forward_unit + time_forward_unit
    travel_rate_backward = 6
    travel_rate_backward_unit = Counter({"miles": 1, "hours": -1})
    time_backward = distance_forward / travel_rate_backward
    time_backward_unit = Counter({"hours": 1})
    assert time_backward_unit == distance_forward_unit - travel_rate_backward_unit
    result = time_backward
    return result
```

=> Example 52 in test dataset.

### A.5.2. INCORRECT OUTPUTS

The challenge of UCPs lies in the requirement to generate accurate `Counter` and `assert` statements. Any program with incorrect `Counter` or `assert` statements is immediately deemed incorrect due to the assertion errors that are raised, leading to decreased performance.

```
def solution():
    """Two cups of flour are needed to make a dozen cookies. Carla is making 36 cookies today and 30 cookies
        tomorrow. How many cups of flour will Carla need to bake the cookies today and tomorrow?"""
    cups_per_dozen_cookies = 2
    cups_per_dozen_cookies_unit = Counter("cups": 1, "dozen": -1)
    cookies_today = 36
    cookies_today_unit = Counter({"cookies": 1})
    cookies_tomorrow = 30
    cookies_tomorrow_unit = Counter({"cookies": 1})
    total_cookies = cookies_today + cookies_tomorrow
    total_cookies_unit = Counter({"cookies": 1})
```

```
        assert total_cookies_unit == cookies_today_unit == cookies_tomorrow_unit
        dozens_of_cookies = total_cookies / 12
        dozens_of_cookies_unit = Counter({"dozens": 1})
        assert dozens_of_cookies_unit == total_cookies_unit - Counter({"cookies": 1, "dozens": -1})
        total_cups_needed = dozens_of_cookies * cups_per_dozen_cookies
        total_cups_needed_unit = Counter({"cups": 1})
        assert total_cups_needed_unit == dozens_of_cookies_unit + cups_per_dozen_cookies_unit
        result = total_cups_needed
        return result
```

=> Example 261 in test dataset. In this case the variable cups_per_dozen_cookies_unit should have a Counter({"cups": 1, "dozens": -1}).

```
def solution():
    """Ashley has an internet connection speed of 20kb per second. Knowing that 1 Mb has 1000 kb, she wants to know
        her internet connection speed in MB per hour. What is Ashley's internet connection speed in Mb per hour?"""
    speed_kb_per_second = 20
    speed_kb_per_second_unit = Counter({"kb": 1, "seconds": -1})
    kb_per_second_to_kb_per_hour = 3600
    kb_per_second_to_kb_per_hour_unit = Counter("kb": 1, "seconds": -1, "hours": -1)
    speed_kb_per_hour = speed_kb_per_second * kb_per_second_to_kb_per_hour
    speed_kb_per_hour_unit = Counter({"kb": 1, "hours": -1})
    assert speed_kb_per_hour_unit == speed_kb_per_second_unit + kb_per_second_to_kb_per_hour_unit
    kb_per_mb = 1000
    kb_per_mb_unit = Counter({"kb": 1, "mb": -1})
    speed_mb_per_hour = speed_kb_per_hour / kb_per_mb
    speed_mb_per_hour_unit = Counter({"mb": 1, "hours": -1})
    assert speed_mb_per_hour_unit == speed_kb_per_hour_unit - kb_per_mb_unit
    result = speed_mb_per_hour
    return result
```

=> Example 627 in test dataset. In this case the variable kb_per_second_to_kb_per_hour_unit should have a Counter({"seconds": 1, "hours": -1}).

```
def solution():
    """Milo is making a mosaic with chips of glass. It takes twelve glass chips to make every square inch of the
        mosaic. A bag of glass chips holds 72 chips. Milo wants his mosaic to be three inches tall. If he has two
        bags of glass chips, how many inches long can he make his mosaic?"""
    chips_per_square_inch = 12
    chips_per_square_inch_unit = Counter("chips": 1, "square inches": -1)
    chips_per_bag = 72
    chips_per_bag_unit = Counter({"chips": 1, "bags": -1})
    bag_count = 2
    bag_count_unit = Counter({"bags": 1})
    total_chips = chips_per_bag * bag_count
    total_chips_unit = Counter({"chips": 1})
    assert total_chips_unit == chips_per_bag_unit + bag_count_unit
    total_square_inches = total_chips / chips_per_square_inch
    total_square_inches_unit = Counter("square inches": 1)
    assert total_square_inches_unit == total_chips_unit - chips_per_square_inch_unit
    mosaic_height_inches = 3
    mosaic_height_inches_unit = Counter({"inches": 1})
    mosaic_width_inches = total_square_inches / mosaic_height_inches
    mosaic_width_inches_unit = Counter({"inches": 1})
    assert mosaic_width_inches_unit == total_square_inches_unit - mosaic_height_inches_unit
    result = mosaic_width_inches
    return result
```

=> Example 808 in the test dataset. In this case the variable chips_per_square_inch_unit should have a Counter({"chips": 1, "inches": -2}) and total_square_inches_unit should have a Counter({"inches": 2}).

