# OpenReview forum: "VerityMath: Advancing Mathematical Reasoning by Self-Verification Through Unit Consistency"
_ICML.cc/2024/Workshop/AI4MATH — ICML 2024 Workshop AI4MATH Poster_

### Official Review · Reviewer_et4K · 2024-06-11

**Rating:** 4
**Confidence:** 3

**Summary:**

The work tackles the challenging math questions in the GSM8K dataset by creating a new Counter (which holds the "measure unit") and the assert statement to help the model learning the answer and the Counter. This should help the model to improve the mathematical reasoning, at least in theory.
In practice authors show that finetuning the model delivers worse results, and authors investigate the reasons.

**Questions:**

Related to W1.

- Maybe the authors could remove some "easy" asserts from the training dataset?

- Or did the model even learn that these tokens are somewhat important?

- Could we build a customize loss to generate a coherent Counter and assert expression?

Related to W2.

- Did the authors experimented with smaller models and found a "scaling law"? Did you try with, say, 100 or 500 examples only?

**Reasons To Accept:**

S1. The approach is very reasonable, especially for physics elementary problems. For example, if we know that a person walks for 300 meters in 100 seconds and we want to know the average speed (given that speed is defined as meters/seconds), a natural guess would be to divide 300 by 100 just to stick with the right measure units. Similarly, authors choose to define Counter and assert to hint the model and let the model create their own representation with a natural guess.

S2. The paper is clearly written and it's pretty easy to follow the motivation and how they create the UCP dataset.

**Reasons To Reject:**

W1. The analysis section (4.5) lacks of a comprehensive explanation. From the learning of section 4.5, it follows that the model is not learning the Counter/assert syntax and more assert is actually penalizing. With this in mind, I would like to see how to incorporate this information in a new training phase.

W2. The impact of training annotations is somewhat pretty generic. It's clear that the in-domain performance is driven from more examples, but it's not clear if the model is too small or if the dataset is too small. It's also not clear how to augment this dataset, given the task is pretty specific and not AI friendly.

W3. The experiment area is very limited. For example, it would be interesting to see why pretrained models have less gap between multiple and single (table 4) compared to finetuned models. I would also be interested in a task where the unit measure is more relevant (maybe physics?).

---

### Official Review · Reviewer_zgBh · 2024-06-12

**Rating:** 6
**Confidence:** 4

**Summary:**

The paper studies the performance of open-source LLMs (Llama 2, Code Llama, Mistral) on math word problems, highlighting challenges with quantities in multiple units. The authors introduce Unit Consistency Programs (UCPs) to ensure unit consistency during operations. Fine-tuning these models with UCPs led to VerityMath variants, which slightly underperformed compared to non-unit-consistent approaches. The paper includes an error analysis and suggests future improvements.

**Questions:**

Do the authors have any ideas to make it work? or explain why UCPs  slightly underperformed the non-unit-consistent approaches?

**Reasons To Accept:**

- The motivation seems sound
- The paper is well-written

**Reasons To Reject:**

- The experimental results do not support the effectiveness of the method.

---

### Official Review · Reviewer_Rxs3 · 2024-06-12

**Rating:** 7
**Confidence:** 5

**Summary:**

This paper introduces VerityMath, an innovative approach aimed at enhancing mathematical reasoning and problem-solving capabilities by emphasizing the self-verification of mathematical expressions through unit consistency. The primary contribution of this work is the development of a framework that allows for the automatic verification of units within mathematical equations, ensuring their consistency and correctness. Additionally, a dataset has been made open source for future research.

**Questions:**

1. Can you compare with more baseline models?
2. Why is the accuracy of the paper method based on GPT-3.5-Turbo and GPT-4 not included in Table 4?

**Reasons To Accept:**

1. The paper introduces VerityMath, an innovative framework that enhances mathematical reasoning through the self-verification of units in mathematical expressions. This concept is not only novel but also highly relevant in verification method where understanding and correctness of mathematical reasoning are crucial.
2. By making their dataset open source, the authors have contributed to the academic community, allowing for replication, further testing, and enhancements by other researchers.
3. The paper is well-written, with clear arguments supported by concrete examples and preliminary testing. The results indicate the framework's effectiveness in achieving its intended goals, making the claims credible and well-founded.

**Reasons To Reject:**

1. Unfortunately, there is no comparison with other verification baseline models, such as [1].
2. What I am more curious about is whether this kind of verification can calibrate the confidence of the model. Some simple experiments can be carried out, such as self-consistency. Maybe this method can break through the upper bound of the voting method.


**Reference**
[1] Weng Y, Zhu M, Xia F, et al. Large language models are better reasoners with self-verification[J]. arXiv preprint arXiv:2212.09561, 2022.
[2] Wang X, Wei J, Schuurmans D, et al. Self-consistency improves chain of thought reasoning in language models[J]. arXiv preprint arXiv:2203.11171, 2022.

---

### Meta-Review · Area_Chair_jzrH · 2024-06-13

**Recommendation:** Accept (Poster)
**Confidence:** 4

**Metareview:**

Acceptable paper. It presents a novel self-verification mechanism of mathematical expressions through unit consistency. Despite not obtaining positive experimental results, the proposed idea is novel, and the contributed dataset is useful for further research. Reviewer et4K has made some good points about the weaknesses of the paper, but overall, I think this paper makes a significant contribution and is suitable for presentation in the workshop.

---

### Decision · Program_Chairs · 2024-06-13

Accept (Poster)